# Defending against black-box adversarial attacks with gradient-free trained sign activation neural networks

## Abstract

While machine learning models today can achieve high accuracies on classification tasks, they can be deceived by minor imperceptible distortions to the data. These are known as adversarial attacks and can be lethal in the black-box setting which does not require knowledge of the target model type or its parameters. Binary neural networks that have sign activation and are trained with gradient descent have been shown to be harder to attack than conventional sigmoid activation networks but their improvements are marginal. We instead train sign activation networks with a novel gradient-free stochastic coordinate descent algorithm and propose an ensemble of such networks as a defense model. We evaluate the robustness of our model (a hard problem in itself) on image, text, and medical ECG data and find it to be more robust than ensembles of binary, full precision, and convolutional neural networks, and than random forests while attaining comparable clean test accuracy. In order to explain our model's robustness we show that an adversary targeting a single network in our ensemble fails to attack (and thus non-transferable to) other networks in the ensemble. Thus a datapoint requires a large distortion to fool the majority of networks in our ensemble and is likely to be detected in advance. This property of non-transferability arises naturally from the non-convexity of sign activation networks and randomization in our gradient-free training algorithm without any adversarial defense effort.

## 1 Introduction

State of the art machine learning algorithms can achieve high accuracies in classification tasks but misclassify minor perturbations in the data known as as adversarial attacks Goodfellow et al. (2015); Papernot et al. (2016b); Kurakin et al. (2016); Carlini & Wagner (2017); Brendel et al. (2018). Adversarial examples have been shown to transfer across models which makes it possible to perform transfer-based (substitute model) black box attacks Papernot et al. (2016a). To counter adversarial attacks many defense methods been proposed with adversarial training being the most popular Szegedy et al. (2014); Tramèr et al. (2018). However this tends to lower accuracy on clean test data that has no perturbations Raghunathan et al. (2019); Zhang et al. (2019) and can still be attacked with better transfer based methods Wu et al. (2020); Xie et al. (2019a); Dong et al. (2019). Many previously proposed defenses have also been shown to be vulnerable Carlini & Wagner (2017); Athalye et al. (2018); Ghiasi et al. (2020) thus leaving adversarial robustness an open problem in machine learning.

A more lethal and practical attack than substitute model training is a boundary based one that requires only the prediction of the model Brendel et al. (2018). These attacks are aimed at finding the minimum distortion to an image such that it will fool a classifier. This is in fact an NP-hard problem for ReLu activated neural networks Katz et al. (2017); Sinha et al. (2018) and tree ensemble classifiers Kantchelian et al. (2016). Even approximating the minimum distortion for ReLu activated neural networks is NP-hard Weng et al. (2018). Boundary based black box attacks such as HopSkipJump Chen et al., Boundary Attack Brendel et al. (2018) and RayS Chen & Gu (2020) give an upper bound on the minimum adversarial distortion.

Binary neural networks that have sign activation and binary weights were originally proposed as lightweight models. These are trained with gradient descent by approximating the sign activation. Recent work has shown that they tend to be more adversarially robust than full precision networks but the improvements are marginal (see Tables 4 and 5 in Galloway et al. (2018) and Table 8 in Panda et al. (2019)).

In this paper we propose a gradient free stochastic coordinate descent algorithm for training sign activation networks with and without binary weights similar to recent work Xue et al. (2020a;b); Xie et al. (2019b). While our original intention was to study the accuracy of a sign activation network trained directly without any approximation we make an interesting finding on the adversarial robustness of our model. We find that ensembling our model gives a high minimum distortion (as measured by HopSkipJump) compared to full precision, binary, and convolutional neural networks. We explain this phenomena by measuring the transferability between networks in an ensemble.

In summary we make the following observations in our paper:

- Our single hidden layer sign activation network has higher minimum distortion than ensembles of full precision and binary neural networks, than random forests that have the advantage of bootstrapping and random feature selection, and than ensembles of convolutional networks that have the advantage of convolutions and several layers.

- Our model's robustness stems from non-transferability of adversarial examples between networks in our ensemble and its robustness increases as we add more networks to the ensemble.

- Substitute model black box attacks require a much greater distortion to bring our model to zero adversarial accuracy compared to ensembles of full precision and binary networks.

- Text classification black box attacks are less effective on our model than on convolutional networks, random forests, and ensembles of full precision and binary networks.

- In a medical diagnosis setting, attacks on ECG data on our model have a higher distortions and are visually distinguishable compared to attacks on ensembles of full precision and convolutional networks, and on random forests.

## 2 METHODS

### 2.1 GRADIENT-FREE STOCHASTIC COORDINATE DECENT

Suppose we are given binary class data $x_i \in R^d$ and $y_i \in \{-1, +1\}$ for $i = 0...n - 1$. Consider the objective function of a single hidden layer neural network with sign activation and 01 loss given below. We employ a stochastic coordinate descent algorithm shown in Algorithm 1 (similar to recent work Xue et al. (2020a;b); Xie et al. (2019b)) to minimize this objective.

$$\frac{1}{2n} \underset{W, W_0, w, w_0}{\arg \min} \sum_i (1 - sign(y_i(w^T(sign(W^T x_i + W_0)) + w_0))) \tag{1}$$

We can train sign activation networks with and without binary weights using our SCD training procedure above. In the case of binary weights we don't need a learning rate. We apply GPU parallelism to simultaneously update features and other heuristics to speed up runtimes (with additional details given in the Supplementary Material).

### 2.2 IMPLEMENTATION, TEST ACCURACY, AND RUNTIME

We implement our training procedure in Python, numpy, and Pytorch Paszke et al. (2019) and make our code freely available from `https://github.com/zero-one-loss/scd_github`. We train three types of sign activation networks with our algorithm: (1) SCD01: 01-loss in the final node, (2) SCDCE: cross-entropy loss in the final node, and (3) SCDCEBNN: cross-entropy in the final node with binary weights throughout the model. Since sign activation is non-convex and our training starts from a different random initialization we run it a 100 times and output the majority vote.

---

**Algorithm 1** Stochastic coordinate descent for single hidden layer network

---

**Procedure:**

1. Initialize all network weights $W, w$ to random values from the Normal distribution $N(0, 1)$.

2. Set network thresholds $W_0$ to the median projection value on their corresponding weight vectors and $w_0$ to the projection value that minimizes our network objective.

**while** $i < epochs$ **do**

    1. Randomly sample a batch of data equally from each class. (We set this to 75% of the training data in image and text data experiments and 25% in the ECG data.)

    2. Perform coordinate descent separately first on the final node $w$ and then a randomly selected hidden node $u$ (a random column from the hidden layer weight matrix $W$)

    3. Suppose we are performing coordinate descent on node $w$. We select a random set of features (coordinates) from $w$ called $F$. For each feature $w_i \in F$ we add/subtract a learning rate $\eta$ and then determine the $w_0$ that optimizes the loss (done in parallel on a GPU). We consider all possible values of $w_0 = \frac{w^T x_i + w^T x_{i+1}}{2}$ for $i = 0...n - 2$ and select the one that minimizes the loss (also performed in parallel on a GPU).

    4. After making the update above we evaluate the loss on the full dataset (performed on a GPU for parallel speedups) and accept the change if it improves the loss.

**end while**

---

To illustrate our real runtimes and clean test accuracies we compare our models with a single hidden layer of 20 nodes to the equivalent network with sigmoid activation and logistic loss (denoted as MLP) and the binary neural network (denoted as BNN) Hubara et al. (2016). We used the MLPClassifier in scikit-learn Pedregosa et al. (2011) to implement MLP and the Larq library Geiger & Team (2020) with the *approx* approximation to the sign activation. This has shown to achieve a higher test accuracy than the original straight through estimator (STE) of the sign activation Liu et al. (2018b).

We perform a 1000 iterations of SCD01 and SCDCE and 10000 of SCDCEBNN. In Table 1 we show the runtimes of a single run of all models on CIFAR10 Krizhevsky (2009) ($32 \times 32 \times 3$, 10K train, 2K test), CelebA facial attributes black hair vs brown hair Liu et al. (2015) ($96 \times 96 \times 3$, 1K train, 1K test), GTSRB street sign recognition 60 vs 120 speed limit signs Stallkamp et al. (2011) ($48 \times 48 \times 3$, 2816 train, 900 test), and ImageNet class 0 vs. 1 Russakovsky et al. (2015) ($256 \times 256 \times 3$, 2580 train, 100 test). Our training runtimes are comparable to gradient descent in MLP and BNN and thus practically usable. We can trivially parallelize training an ensemble by doing multiple runs on CPU and GPU cores at the same time. We also show test accuracies of 100 vote ensembles of all models and find our model accuracies to be comparable to MLP and BNN.

Table 1: Training runtimes of single run in seconds and test accuracies of 100 vote ensembles in parenthesis for binary classification

|  | **SCD01** | **SCDCE** | **SCDCEBNN** | **MLP** | **BNN** |
|---|---|---|---|---|---|
| CIFAR10 | 64 (87%) | 56 (88%) | 422 (87%) | 13 (90%) | 106 (83%) |
| CelebA | 20 (79%) | 18 (81%) | 111 (72%) | 41 (78%) | 32 (76%) |
| GTSRB | 22 (97%) | 22 (97%) | 92 (98%) | 8 (99%) | 42 (96%) |
| ImageNet | 77 (72%) | 54 (73%) | 338 (71%) | 115 (72%) | 78 (66%) |

## 3 RESULTS

Going forward we compare the adversarial robustness of ensembles of our three models SCD01, SCDCE, and SCDCEBNN, their full precision and binary gradient descent trained equivalent counterparts MLP and BNN, two convolutional neural networks: LeNet LeCun et al. (1998) and ResNet50 He et al. (2016), and random forests Breiman (2001) (denoted as RF). For each model we use the majority vote output of 100 votes each with different initial parameters except for ResNet50 where we use 10 votes. In random forest we use an ensemble of 100 trees.

We use a single hidden layer of 20 nodes in our three models and in MLP and BNN throughout the paper. The convolutional networks and random forest are not a fair comparison to our model since it has fewer parameters and does not perform bootstrapping or random feature selection as random forest. We include them nevertheless since convolutional neural networks serve as state of the art references and random forest serves as an alternative ensemble method.

## 3.1 Adversarial distortion on image data

The minimum distortion required to make a datapoint adversarial is an indicator of a model's adversarial and even corruption/general robustness Gilmer et al. (2019). We consider 10 randomly selected datapoints from the CIFAR10 benchmark Krizhevsky (2009) and report their minimal adversarial distortion as given by HopSkipJump Chen et al., Boundary Attack Brendel et al. (2018) and RayS Chen & Gu (2020).

We use the HopSkipJump and Boundary Attack implementation in the IBM Adversarial Robustness Toolkit (ART) Nicolae et al. (2018) In order to obtain as accurate an estimate as possible we run both mthods 10 times each with an initial pool size of 1000 random datapoints and maximum iterations of 100 and report the minimum value. For a single datapoint this typically takes several hours to finish and thus we are able to report the distortion of only 10 random points in this study. We use the RayS implementation from their GitHub site `https://github.com/uclaml/RayS` and run it with default parameters of 40,000 queries to obtain a distortion estimate.

In Table 2 first row we show the clean test accuracy of all models on CIFAR10 class 0 vs. 1. The convolutional networks LeNet and ResNet50 have higher accuracies since they have the advantage of convolutions. In the following three rows of Table 2 we see the minimum adversarial distortion of models as estimate by three boundary attack methods. We were unable to attack some models with Boundary Attack and RayS due to time constraints and mark them as NA. We see that HopSkipJump gives the lowest distortion for each model except for SCD01 and SCDCE where it is comparable to RayS.

Amongst HopSkipJump distortions our sign activation trained models have the highest adversarial distortion with the binary weights cross-entropy variant as the winner. All other neural networks lag far behind and have distortion even lower than random forest. Even though BNN also has sign activations its distortions are similar to MLP possibly due to its approximation of the sign activation and gradient descent search. If we use the the straight through estimator and swish approximations Darabi et al. (2018) the distortions remain similar to what we report here.

Table 2: Mean minimum $L_2$ distortion of 10 random test images from CIFAR10 class 0 vs. 1 as estimated by three different boundary attack methods. Highest distortion by HopSkipJump shown in bold.

|  | SCD01 | SCDCE | SCDCEBNN | MLP | BNN | ResNet50 | LeNet | RF |
|---|---|---|---|---|---|---|---|---|
| Clean acc | 87 | 88 | 88 | 90 | 83 | 98 | 96 | 88 |
| HSJ | 3.2 | 3.36 | **3.6** | 0.77 | 0.76 | 0.76 | 1.73 | 1.91 |
| Boundary | 7.69 | 8.23 | NA | 2.47 | NA | 3.44 | 7.29 | 6.63 |
| RayS | 3.14 | 3.08 | NA | 0.99 | NA | NA | 2.54 | 6.77 |

To further validate the distortions above we run HopSkipJump on SCD01, MLP, LeNet, and RF with 10 maximum iterations on the first 100 CIFAR10 test datapoints. We used a fixed image as the initial one in these experiments. In Table 3 we see that SCD01 distortions are the highest and the relative ranking is the same as we saw for the 10 images above with 100 maximum iterations of HopSkipJump.

In Table 4 below we show HopSkipJump distortions (min of 10 runs 100 max iterations each) on a single random image from CelebA, GTSRB, and ImageNet datasets. We find our SCD models to have a higher distortion on both CelebA and GTSRB but comparable to MLP on ImageNet.

To illustrate our model's scalability we show HopSkipJump distortion values for our SCD01 model with different number of hidden nodes.

Table 3: Mean minimum $L_2$ distortion of first 100 test images from CIFAR10 class 0 vs. 1 as estimated by HopSkipJump with 10 maximum iterations starting from fixed initial images. Highest distortion in bold.

| | SCD01 | MLP | LeNet | RF |
|---|---|---|---|---|
| HSJ | **6.25** | 0.83 | 2.98 | 4.64 |

Table 4: Minimum $L_2$ adversarial distortion of a single random test image from CelebA, GTSRB, and ImageNet class 0 vs. 1. In bold are the largest distortion values for each dataset.

| | **Celeba** | | | | | | | |
|---|---|---|---|---|---|---|---|---|
| | SCD01 | SCDCE | SCDCEBNN | MLP | BNN | ResNet50 | LeNet | RF |
| Image 0 | 8.77 | 8.6 | **14.13** | 1.02 | .22 | 1.68 | 3.3 | 2.82 |
| | **GTSRB** | | | | | | | |
| | SCD01 | SCDCE | SCDCEBNN | MLP | BNN | LeNet | RF | |
| Image 0 | .6 | .82 | **1.24** | .62 | .87 | .33 | .01 | |
| | **ImageNet** | | | | | | | |
| | SCD01 | SCDCE | SCDCEBNN | MLP | BNN | ResNet50 | RF | |
| Image 0 | 20.9 | 16.17 | 3.26 | **24.1** | 5.68 | 2.01 | 5.78 | |

Table 5: Mean minimum $L_2$ distortion of a single test image from CIFAR10 class 0 vs. 1 for different number of hidden nodes in our SCD01 ensemble model. Highest distortion in bold.

| | **SCD01** | | | | |
|---|---|---|---|---|---|
| Hidden nodes | 4 | 16 | 20 | 32 | 64 |
| Distortion | 2.22 | 1.98 | 2.21 | 2.27 | **3.45** |

## 3.2 TRANSFERABILITY WITHIN ENSEMBLES AND EFFECT OF ENSEMBLE SIZE

To understand the above phenomena we estimate the probability that an adversarial example targeting a single model in the ensemble will also be adversarial to other models in the ensemble. We can estimate this by first performing a HopSkipJump attack on each model in the ensemble separately. Let $x_i'$ be the adversary obtained by targeting model $m_i$ in the ensemble. Let $k_i$ be the number of models in the ensemble that are also misclassified by the adversary $x_i'$ (thus transferable). We sum $k_i$ for $i = 0...n-1$ and divide by 9900 which is the maximum value of this sum (obtained when the adversary attacks all models in the ensemble excluding the target of course).

We average this probability for Images 0 through 7 for each method. In Table 6 we see that this probability is lowest for our models and highest for MLP and BNN. The fact that this probability is very low for our models indicates that for several of the networks in our ensemble the adversary targeting a fixed network does not transfer to other ones. The low transferability of our models indicates that a greater distortion is required for an image to be adversarial.

Table 6: Estimated probability that an adversarial image targeting a single model in the ensemble (of 100 models) will transfer to other models. Lowest probability in bold.

| | SCD01 | SCDCE | SCDCEBNN | MLP | BNN | ResNet50 | LeNet | RF |
|---|---|---|---|---|---|---|---|---|
| Prob | .006 | .004 | **.002** | .39 | .2 | .02 | .01 | .07 |

In fact as we see in Figure 1 the robustness of our models increases as we increase the ensemble size to a much larger degree than ensembles of MLP and BNN, and than RF. We use ensemble sizes of

100 in this study but the figure suggests that increasing our ensemble size is likely to further increase robustness.

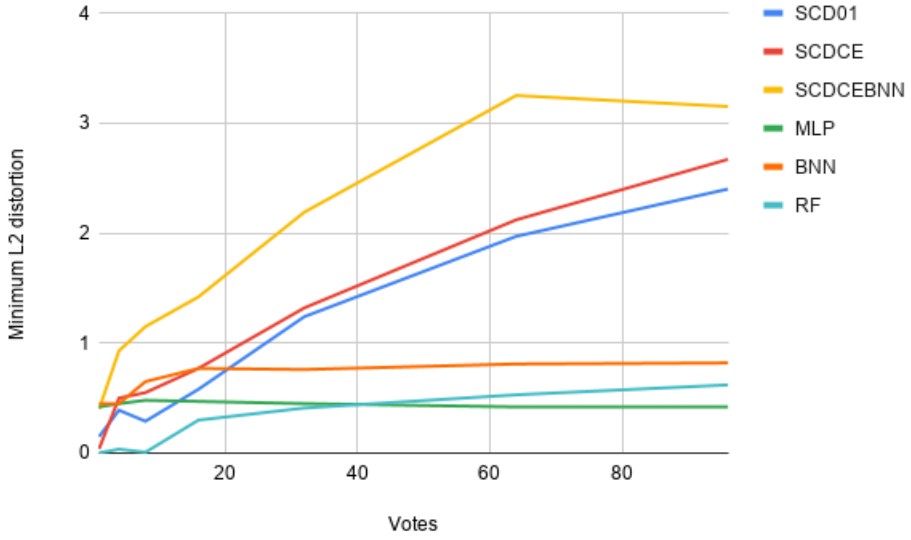

Figure 1: Minimum $L_2$ image distortion as a function of ensemble size

### 3.3 SUBSTITUTE MODEL BLACK-BOX ATTACKS

Our model's high distortion in CIFAR10 is reflected in substitute model black box attacks on this dataset Papernot et al. (2016a). We train a two hidden layer neural network each with 200 nodes as the substitute using the standard adversarial augmented algorithm Papernot et al. (2017) (described fully in the Supplementary Material). In Figure 2 we that our models require a much higher distortion than their gradient descent trained equivalents MLP and BNN in order to reach zero percent adversarial accuracy. We also see that all models attacked with random Gaussian noise of the same distortion added to the test examples are barely affected thus showing the effectiveness of the black box adversarial examples.

### 3.4 TEXT BLACK-BOX ATTACKS

The TextFooler Jin et al. (2020) method is designed to find syntactically and semantically similar adversarial documents by replacing important words with similar ones until the document is misclassified. We apply this to all ensemble models on four document classification datasets: Internet Movie Database (25K train, 25K test, mean words per document: 215) and Yelp (560K train, 38K test, mean words per document: 152) positive and negative reviews (IMDB and Yelp), sentence classification of positive and negative sentiments (9K train, 1K test, mean words per document: 20, denoted as MR), and sentence-level classification of news items in World and Sports categories (120K train, 7.6K test, mean words per document: 43, denoted as AG) Jin et al. (2020).

WordCNN stacks word vectors Pennington et al. (2014) of each word in a document into a matrix to treat it as 2D image Kim (2014). In the other models that take feature vectors as inputs we consider the averaged word vector of all words in a document Lilleberg et al. (2015). For all models we use 200 dimensional Glove word embeddings pre-trained on 6 billion tokens from Wikipedia and Gigawords Pennington et al. (2014). This gives a lower clean test accuracy than WordCNN but still above an acceptable level in practice.

In Table 7 we see that ensembles of our models give the highest adversarial accuracy on all four datasets and require the greatest number of queries. If a smaller limit was placed on the allowed queries (for example imposed by the system being attacked) we can expect a higher adversarial

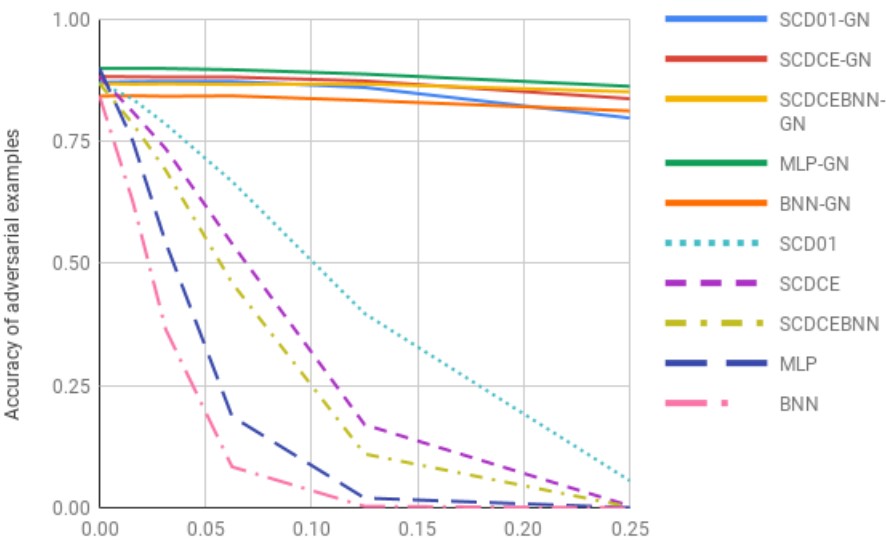

Figure 2: Accuracy of adversarial examples and test data with Gaussian noise (denoted as -GN for each model) for various distortion thresholds on CIFAR10. The substitute model adversarial examples are far more effective than random noise. At distortion 0.125 both MLP and BNN have near 0% accuracy whereas SCD01 has 40%.

accuracy for our models. Here we show ensembles of 8 votes for each model. If we increase the ensemble size to 100 we find the adversarial accuracy of our models, BNN, and RF slightly drop but their relative difference remains the same.

Table 7: Accuracy of clean and TextFooler black-box adversarial examples denoted by Cl and Adv respectively. All models shown here are 8 votes instead of 100 as we did for CIFAR10 images above. Also shown are the number of queries made by the attacker denoted as Que. We round accuracies and queries to the nearest integer. For each dataset highest adversarial accuracy shown in bold.

|  | **IMDB** | | | **Yelp** | | | **MR** | | | **AG** | | |
|  | **Cl** | **Adv** | **Que** | **Cl** | **Adv** | **Que** | **Cl** | **Adv** | **Que** | **Cl** | **Adv** | **Que** |
|---|---|---|---|---|---|---|---|---|---|---|---|---|
| SCD01 | 82 | 51 | 3279 | 85 | 54 | 1908 | 74 | 14 | 186 | 99 | 93 | 672 |
| SCDCE | 84.8 | **52.4** | 3255 | 86 | **56.7** | 1903 | 75.8 | **16.7** | 189 | 99 | **94.5** | 687 |
| CNN | 89.2 | 0 | 524 | 94 | 1.1 | 492 | 78 | 2.8 | 123 | 96.5 | 49.1 | 258 |
| MLP | 85 | 0 | 686 | 87.3 | . 2 | 500 | 75 | 2.3 | 123 | 99 | 51.4 | 366 |
| BNN | 83.8 | 21.5 | 2301 | 85 | 32.3 | 1622 | 73.2 | 5.8 | 150 | 99.1 | 75.6 | 564 |
| RF | 76.7 | 11 | 1823 | 77.7 | 7.5 | 935 | 68.1 | 2.1 | 115 | 96.7 | 72.2 | 532 |

## 3.5 ECG BLACK-BOX ATTACKS

ECG time-series data is increasingly being used in automatic diagnosis by machine learning systems Ribeiro et al. (2020). Tailored adversarial attacks have recently been proposed Han et al. (2020); Chen et al. (2020) but HopSkipJump can also be used to produce adversarial ECG examples. To illustrate this and evaluate our model's robustness on this data we consider the PTB Diagnostic ECG dataset Bousseljot et al. (1995); Goldberger et al. (2000) available from this URL `https://www.kaggle.com/shayanfazeli/heartbeat`. We randomly split this dataset into an 80:20 train test split (yielding 13096 train and 1456 test points).

We train 100-model ensembles of SCD01, SCDCE, and MLP. We also train an ensemble of 10 convolutional neural networks (CNN) with 1D convolutional kernels, random forest (RF) with 100 trees, and a 10-model ensemble of BNN (as opposed to a 100-model ensemble which is slow to attack and did not show a better distortion on selected datapoints). Each of our CNNs has the following structure: 64 1x16 Conv1D kernels $\rightarrow$ MaxPool 1x4 $\rightarrow$ 128 1x16 Conv1D kernels $\rightarrow$ MaxPool 1x4 $\rightarrow$ 256 1x16 Conv1D kernels $\rightarrow$ MaxPool 1x2 $\rightarrow$ FullyConnected $\rightarrow$ Output. In Table 8 we see the clean test accuracy of our model is slightly lower than gradient-descent trained models and random forest. We picked 37 random datapoints from the test and attacked all models on these points. We attack each point 10 times and report the minimum with the same parameters as we did in our CIFAR10 attacks described above.

In Table 8 second row we show the average min $L_2$ distortion and find SCD01 to have the highest one. The $L_2$ difference between SCD01 and the next best RF (after SCDCE) turns out to be statistically significant with a p-value of .008.

Table 8: Clean test accuracy and minimum $L_2$ adversarial distortion on the PTB Diagnostic ECG dataset. Highest distortion shown in bold.

|  | SCD01 | SCDCE | MLP | BNN | CNN | RF |
|---|---|---|---|---|---|---|
| Clean acc | 91.1 | 93.3 | 96.1 | 80.4 | 99.6 | 97.6 |
| Average min $L_2$ distortion | **.19** | .14 | .08 | .14 | .1 | .14 |

In Figure 3 we visualize an original ECG sample and its adversarial versions targeting SCD01, CNN, and RF. The SCD01 adversary is rigid and has many more bumps compared to the CNN and RF adversaries and is thus likely to be detected by an observer or a system that checks for smoothness (that we expect to see as in the original sample).

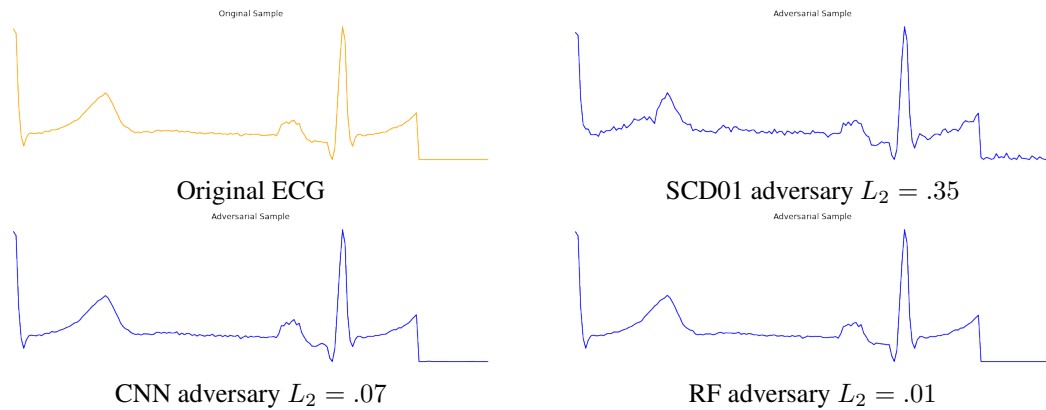

Figure 3: Original and adversarial ECG examples

## 3.6 DISCUSSION

Using ensembles of neural networks and promoting diverse ensembles has been previously proposed as a defense against adversarial attacks. Studies using ensembles with different initializations (like we do), bootstrapping, and Gaussian noise have shown robustness but only in the white box setting Strauss et al. (2017) (which is somewhat unrealistic since it assumes the attacker has full knowledge of the model and its parameters). Other studies combine the loss of all models in the classifier and add a regularizer that promotes diversity.

For example we could try to maximize the angle between gradients of models in the ensemble Kariyappa & Qureshi (2019) to make them *misaligned*. In their diversity training they use a Gaussian noise augmented dataset which raises concerns about the effectiveness of their method since augmentation alone has been shown to be effective in ensemble training Strauss et al. (2017). An-

other study maximizes diversity between classes Pang et al. (2019) and thus does not apply to our work here that focuses on binary classes only. Even for multiple classes their method is computationally expensive as it uses a joint loss function. Other methods inject noise to models in the ensemble Liu et al. (2018a) but their evaluation is only in the white box setting. Various measures for ensemble diversity have been previously proposed for deep networks Liu et al. (2019) and evaluated in the white-box setting.

We can apply all of the above diversity training methods to our ensemble of sign networks. Our work, however, is not explicitly aimed at enhancing diversity. As we show it is naturally diverse and we conjecture this is due to the non-convexity of sign activation and our randomized training method. Even sigmoid activation networks have a non-convex search space but we can imagine that sign activation gives a greater degree of freedom. This can easily be seen in the case of a linear classifier with logistic or hinge loss vs. 01 loss Xue et al. (2020b).

Our model accuracy is not the same as convolutional networks understandably due to lack of convolutions in our networks. But they are close to sigmoid activated networks and random forests, and better than binary neural networks in most cases. It is possible to extend our training procedure to allow for convolutions and this may increase accuracy making our model comparable to convolutional networks and much more robust.

It is hard to make a general claim of robustness with only 100 images from CIFAR10. We would need to show more images from CIFAR10 and other image benchmarks as well but our preliminary experiments on CelebA, GTSRB and ImageNet (shown in Table 4) suggest higher distortion on other image data as well. Due to computational limitations we are unable to show more image data here but instead we take another route to show generality of our results. We show that our model is robust even to text classification black box attacks and on ECG data attacks. Both of these are outside the domain of images and our model's robustness there suggests a greater generalization. Future work entails extending our training to sign activated convolutions and multi-class networks.

## 3.7 CONCLUSION

We show that our ensemble of gradient-free sign activation networks are harder to attack than ensembles of several other networks and random forests on images, text, and medical data.

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
