# OpenReview forum: "Defending against black-box adversarial attacks with gradient-free trained sign activation neural networks"
_ICLR.cc/2021/Conference — Reject_

### Official Review · AnonReviewer4 · 2020-10-24
**Lots of work is needed to improve the paper**

**Rating:** 4
**Confidence:** 4

**Review:**



This paper proposes training an ensemble of binary neural networks with sign activations using gradient-free stochastic coordinate descent algorithm.
The nature of the training method and binary networks leads to robust models with non-transferable attacks.

----
*Pros*:
I appreciated
- the non-transferability experiment which demonstrated that the trained ensemble is diverse and sort-of "orthogonal".
- the use of minimum distortion as a measure of the model's robustness.

*Cons*:
- It looks like the method is not scalable. Experiments were carried out on a single-hidden-layer network of 20 nodes (would really like to see experiments on bigger models)
- Only one black-box attack is used to demonstrate the effectiveness of the method and that too under one perturbation threat (l2-norm). There are multiple black-box attacks that exploit different aspects of model vulnerabilities. It cant be claimed that the method is robust based on a single-attack single-threat.
- The scale of experiments (10 images) does not provide significant evidence that this method is effective and the authors acknowledge that in Section 3.6


Remarks/Comments:
- Page 2, Line2: "by with gradient" -> "with gradient".
- Sec 3.2: "divide by 9900" -> "divide by 99".
- Table 1: What does "single run" mean in the caption? Is it a single forward-pass?
- Fig 2: please make it more legible with markers (colors are not enough).


Based on the above, I suggest that the authors redesign their experiment to support the paper's proposition in terms of the method scalability and robustness against more attacks & perturbation settings. Perhaps making the experiments more scalable by reducing the compute budget of HopSkipJump (1000 random init x 10 times x 100 iterations)

**Post-Rebuttal**: No change in my score, please read my comments in the thread below.

---

> ### Author Response · Authors · 2020-11-25
> **Added more results**
>
> Thank you for your feedback. We have added distortion estimates by two other boundary based black box attack methods and also added distortions on more CIFAR10 datapoints. Our conclusions still hold after these additions.
>
> We can train models with more hidden nodes and even have a model with a single convolutional layer. Due to rebuttal time constraints we are unable to complete distortion results on those models. To give you a sense of our model's scalability we provide HopSkipJump distortion estimates (min of 10 runs maxiters=100) for different hidden nodes in our single hidden layer model (100 vote ensemble):
>
> Hidden-nodes:  4,      16,       20,      32,       64
>
> Distortion:         2.22,  1.98,   2.21,   2.47,    3.45
>
> We expect our model distortion to increase with more hidden nodes and with a convolutional layer.
>
> Response to your comments:
>
> 1. Fixed
> 2. Divide by 9900 is actually correct. We have a total of 100 models m0 through m99 in each ensemble. Consider the adversary x'0 that attacks model m0. The maximum number of models that the adversary x'0 succeeds in attacking by transferability (besides its target m0) is 99. Now consider adversary x'1 that attacks m1. The maximum number of models it can succeed in transferring to is 99. We sum this for each of the adversaries x'0 through x'99 and get the max value of 9900. Thus a small probability means that adversaries transfer to a small fraction of the 9900 max value.
> 3. It means the total training runtime of one instance of our model. We then consider an ensemble of 100 models which means training 100 models separately. Thus the total runtime is the single run times 100 but this can be parallelized.
> 4. We use different line types for the models and also ordered the (non-GN) methods from top to bottom according to their adversarial accuracy in descending rank.
>
> We have added distortions on 100 more CIFAR10 test points by running HopSkipJump with 10 maximum iterations and fixed initial images (10 runs per image since even with a fixed initial image the distortions can vary).

---

### Official Review · AnonReviewer2 · 2020-10-27
**an experimental paper that could be improved**

**Rating:** 5
**Confidence:** 4

**Review:**

The paper proposes an architecture (ensemble of networks) aiming at being robust against black-box attacks, based on the idea that crafting an adversarial example able to fool enough individual networks such that the majority vote changes is a more difficult task.  The paper presents ways of training such ensembles and provides several sets of experiments showing the advantage of the approach. It also contains an observation on "non-transferability", counting how many co-networks are fooled when only one is targetted by the blackbox attack. It turns out that this amount is lower for the proposed scheme.

The algorithms are postponed to supplementary material and the paper itself mainly report the experimental part. It concerns :
-> training time
-> minimum adversarial distorsion (l2 and l_inf)
-> transferabiliy
-> different tasks (images, text and ecg)

All tested algorithms are ensembles and dataset are subparts of actual datasets.

Comments:
* Table 1 : training time of a single run : should not it be avaraged on several runs somehow?
* Table 2, 3, 5 : I would find it more readable if some remarkable values were bold for instance
* Table 4 : since all models don't have the same number of weak learners, I suggest that the presentation of the table recalls it, it * has a great impact on values here
* (a verb is missing on top of page 6. )
Figure 2 : hard to read. Top lines appears to be  some kind of reference to be compared to and the bottom lines the proposed methods. I suggest that one color is assigned to each algorithm and different line style for with or without GN. Also I think that there's a typo in the legend (SCDCE-BNN-> SCDCE-GN?)
* biblio : I did not check each paper on arxiv but I'm quite sure that a significative part of arvix references have some published references.  Citing preprints is ok for recent work only. The reference by Alex Krizhesky seems to lack information too.

Overall, I'm not really convinced by the paper in its present form, although I recognize that the results show some interesting properties for robustness.  Experiment on running time seems a little misleading, and shows that the most efficient variant is also significantly slower. I'm actually not against slower methods, but when I read this part I have a feeling of "is it right?"
I also think that there would be enough space to provide more information on algorithms in the main paper.
The transferability idea seems to be an interesting point, but it has to be further developped. How to apply this observation to the actual blackbox attack, which, as far as I understand, does not attack weak learners one by one?

---

> ### Author Response · Authors · 2020-11-25
> **Added more data to our paper**
>
> Thank you for your feedback. We have tried to make our case more convincing by adding more data. To address your other comments:
>
> 1. The runtime of single runs are similar with low standard deviation, thus we report just one run
> 2. Important values are in bold now
> 3. We now specify in Table 4 caption that we use 8 votes for the text models.
> 4. We have fixed the legend typo in Figure 2 and ordered the legend method according to their adversarial accuracy (methods with high accuracy are shown above the ones with lower accuracy). We have also shown the non-GN methods with different line types for better legibility.
> 5. We have fixed the citations.
>
> Our running times are correct. The SCDCEBNN model performs 10000 epochs whereas the SCD01 and SCDCE has 1000 epochs and thus are much fasters. The most correct variant MLP is fastest because it uses gradient descent which is much faster than our coordinate descent.
>
> We have added a better description of our training algorithm.
>
> The black box attack is performed on the ensemble. Our point is that the networks in our ensemble are non-transferable and this leads to some type of orthogonality that forces an adversary to undergo many changes before it can fool the entire ensemble. In comparison ensembles of other models don't have the same level of non-transferability (orthogonality) as we show in our paper.

---

### Official Review · AnonReviewer1 · 2020-10-28
**Unconvincing evaluation**

**Rating:** 3
**Confidence:** 4

**Review:**

This paper presents a new method to defend black-box attack based on an ensemble of sign activation neural networks. The authors demonstrate their method has much higher minimum distortion using HopSkipJump attack.

However, I have many concerns regarding this paper:

-The paper organization is very weird and confusing. The author did not put their main algorithm into the main paper. Instead, they put many unimportant results (e.g. Table 1) into the main paper. If there is not enough space the author should use simpler sentences to describe their algorithm and put some results into supp. Also, the figure and table style (i.e., unbounded table, screenshot figures) makes me feel this is an undergrad project report instead of an ICLR submission.

-Besides transfer attack, the authors only evaluate the black-block robustness using HopSkipJump attack. Their claim is "Compared to other boundary attack methods it is known to give the best estimate of a datapoint’s minimum adversarial distortion."
Is there any paper support this claim? I don't believe one attack method is universally better than other method among all datasets.
I think the authors should evaluate a set of attack methods instead of only one method otherwise the results are not convincing to me.

-What is the purpose of providing detailed results of 10 random images in Table 2, 3? Those results not only occupied a lot of space but also did not provide any useful insight. An average number of the entire dataset is enough.

-Citing issue: when referencing a paper the author should use the published version not the arxiv version if the cited paper is published.
E.g., Angus Galloway, Graham W Taylor, and Medhat Moussa. Attacking binarized neural networks. arXiv preprint arXiv:1711.00449, 2017.
should be
Galloway, Angus, Graham W. Taylor, and Medhat Moussa. Attacking Binarized Neural Networks. International Conference on Learning Representations. 2018.


-The baseline comparison are all undefended networks. The author should compare to some other blackbox defense methods.

---

> ### Author Response · Authors · 2020-11-25
> **Added more data to make our case convincing**
>
> Thank you for your feedback. We agree that evaluating robustness of a model is a hard task. To make our case convincing we show that HopSkipJump distortions are tighter than Boundary Attack and RayS, both of which also estimate the adversarial distortion. We also show HopSkipJump distortions on more datapoints and find our model to still have the highest distortion.
>
> We have fixed the citations, removed distortions on individual datapoints, and added a clearer description of our training algorithm. We have a also bounded our tables and made the legend of Figure 2 clearer.
>
> To show distortions on adversarially trained models we would have to figure out how to adversarially train our new models. We would then have to evaluate the distortion of the adversarially trained models, both of which are impossible within the rebuttal time limitation. In the original HopSkipJump paper though adversarial training does not seem to improve the model's distortion shown on MNIST only.

---

### Author Response · Authors · 2020-11-25
**Uploaded new rebuttal version**

We have incorporated all reviewer feedback (to which we are thankful) into our revised version. We have also included (single image) distortions on CelebA, GTSRB, and ImageNet, and single image CIFAR10 distortions of our model with different number of hidden nodes.

---

### Decision · Program_Chairs · 2021-01-07
**Final Decision**

**Decision:**

Reject

**Comment:**

The paper proposes an algorithm to defend against black-box attacks. All the reviewers think the current experiments are not convincing enough, and the method seems to have some issues (e.g., not scalable).